# Phenomenon-Based Learning in Teaching a Foreign Language: Experiences of Lithuanian Teachers

Nijole Ciuciulkiene *, Ilona Tandzegolskiene-Bielaglove and Martyna Culadiene

Education Academy, Vytautas Magnus University, 44244 Kaunas, Lithuania;
ilona.tandzegolskiene-bielaglove@vdu.lt (I.T.-B.); martyna.culadiene@stud.vdu.lt (M.C.)
* Correspondence: nijole.ciuciulkiene@vdu.lt

**Abstract:** Phenomenon-based learning (hereinafter PhenoBL) is widely studied in the majority of European countries, especially given that research data indicate that PhenoBL is more successful in providing effective learning, better student achievement, a stronger interest in science, and even a higher happiness index. However, there are sparse data on the educational practice of this method in Lithuania, particularly in foreign language teaching (FLT). Thus, teachers' professional preparation for the effective implementation of PhenoBL remains one of the most relevant research problems. For this reason, this study aims to analyse the experiences of Lithuanian foreign language teachers in incorporating PhenoBL into FLT. Fifteen individual semi-structured interviews were conducted, and the obtained data were analysed by applying qualitative inductive content analysis. An inductive content analysis of the interview reports revealed six themes and related categories: the perception of student-centred teaching, the development of subject integration competencies, teamwork development competencies, research-planning skills, the positioning of personal responsibilities and duties, and foreign language usage emancipation, i.e. setting free from personal fears (fear to make grammar, vocabulary mistakes, while speaking in public) to speak a foreign languages. The content of the revealed themes indicated that teachers highlighted the flexibility of PhenoBL from the perspective of its application to different language learning levels within one group. The majority of the respondents underlined the necessity for the development of an active didactic competence. Other respondents mentioned the importance of the correlation between personal creativity competence development and success in PhenoBL. It was also stressed that if a teacher wants to be successful while using PhenoB, they must to be prepared to work with integration-based and communication-emancipatory methods, must be student-centred, must have competencies in teaching several subjects, must be good at teamwork, and must be good at managing learning time.

**Keywords:** phenomenon-based learning (PhenoBL); teaching a foreign language; didactic competencies of foreign language teachers

## 1. Introduction

The new requirements and approaches for the future standards and competencies that students should demonstrate are changing the attitudes about the basic skills and knowledge expectations that dominated in the past. More and more research is being directed towards the investigation of key soft-skill competencies, known as the "4Cs": creativity, critical thinking, collaboration, and communication (Thornhill-Miller et al. 2023). For this reason, schools must be transformed in a way that will enable students to acquire multitasking skills, become multiliterate, demonstrate the ability to think sophisticatedly, flexibly solve emerging problems, and collaborate and communicate in a way that promotes their success in work and life.

The majority of scientists agree that PhenoBL is directed towards the development of the "4Cs" while investigating certain real-world phenomena from different perspectives (Meriläinen and Piispanen 2012; Symeonidis and Schwarz 2016; Makarova et al. 2020;

Akkas and Eker 2021). The researchers mentioned here highlight the idea that PhenoBL truly stands for relevant, innovative, substantive, and timely learning. This concept introduces PhenoBL not only as a teaching method, but also as a new way of thinking (Roiha and Polso 2021).

This new vision arose from students' curiosity, self-motivation, autonomy, and individual efforts to pursue and explain the holistic, real-world phenomena around them. In this context, holistic, real-world phenomena relate to practical and realistic life topics (not simulated, but real-life situations), such as human relations, media and technology, natural resources, and other related socio-cultural issues that could provide and enliven teaching and learning objectives in the pedagogical environment so that they become accessible, concrete, and meaningful to learners (Roiha and Sommier 2018; Akkas and Eker 2021; Silander et al. 2022).

This concept also refers to redesigning teaching so that learning can take place in problem-solving contexts. Such contexts provide learners with the possibility of plunging into language usage while being constantly encouraged to actively participate in investigating and analysing specific academic information, reflecting on emerging challenges with peers and groups, debating and negotiating, obtaining results, making conclusions, and reflecting on their experiences throughout the learning process (Roiha and Polso 2021).

The introduction of phenomenon-based learning (PhenoBL) might help to solve this task in a positive way (Binkley et al. 2011; Randolph et al. 2009). PhenoBL is widely studied in European countries, especially in Finland. In 2014, the Finnish education system made a progressive decision; starting from 2016, it would include PhenoBL in the curricula of comprehensive schools. The essence of the Finnish education reform was to change the "what?" and "why?" of learning into one simple "how". This may be regarded as a response to critical statements proposing that traditional schools present too much theoretical and fragmented learning instead of linking it to real-world issues and problems (Kangas and Rasi 2021).

PhenoBL, like other learning processes—i.e., problem-based learning, inquiry-based learning, and project-based learning, different from content-based and task-based learning—is designed to transform the goal of learning into a process, not just a result. This means not only learning facts but being able to apply them (Kangas and Rasi 2021; Bărbuleţ 2022). While implementing this princip, teachers turn towards active learning. The whole learning process is based on investigating real-life phenomena. Teachers plan and compile learning tasks that enable students to create their individual approaches and answers as students have a possibility to relate them to their personal experiences. When the content of the lesson is connected to real life, the students are able to participate more actively in learning as the whole process becomes more close and clear to them (Johnson 2021).

During classes, PhenoBL proved to be an effective and reliable strategy for responding to pupils' needs, promoting their motivation to learn (particularly in the teaching of foreign languages, as it encourages verbal communication) since the topics were presented according to their interests, and stimulating their language skill development and cultural awareness developed their communicative skills, improved the level of language acquisition the evidence of which is students' personally formed active linguistic and communicative thesauri, which might be regarded as compendiums of relevant bodies of knowledge with the structured relationship of concepts within an application of given thematic area(Meriläinen and Piispanen 2012; Symeonidis and Schwarz 2016; Lonka et al. 2018). PhenoBL is based on questioning and problem-solving skills (Akkas and Eker 2021; Kangas and Rasi 2021).

At present, English is accepted as the leading lingua franca in the world. More and more people around the world are choosing to learn English as their main foreign language. Alongside the dominant communicative function of English, initiatives for learning other foreign languages are being developed. Thus, current foreign language teachers are facing the task of finding the most effective strategies to ensure success in every student's foreign language acquisition, ensuring that a foreign language might become a tool and helping them to be successful in the global socialisation process (Protassova 2018; Johnson 2021).

The combining of language learning with phenomenon-based learning is rather common. Instead of organising separate classes for each subject, students take part in cooperative project-based learning units/activities that include content standards and objectives from all main curriculum subject areas. These projects not only enable students to learn about specific topics in their own way but also make learning more meaningful. Instead of students being asked to memorise facts for the sole purpose of completing an exam, students learn how to solve problems in a multifaceted way (Johnson 2021), realising all four language domains: listening, reading, speaking, and writing. In other words, during PhenoBL sessions, the students are engaged in active communication in a foreign language process, both in written and oral forms.

The varieties of foreign language usage are organised through six stages of PhenoBL: (1) Establishing the Phenomena Goals; (2) Engaging and Immersing the Learners; (3) Personalised Learning; (4) Establishing Evidence-Based Inquiry to Knowledge; (5) Questioning for Reflection; and (6) Documenting a Portfolio and Sharing the Project (Varzari 2022). PhenoBL class teachers should discuss the project design and tasks with the students. The learning process design should highlight the fluent stages: designing, choosing different learning tasks, selecting practical activities, and considering ways of guiding and assessing (Varzari 2022).

The essential crux of PhenoBL is the opportunity for each participant to become actively involved in communication regarding the content studied, performance, and results, as well as what happens around them. A teacher's main concern is to monitor the discovery–learning process by facilitating students' independent decoding of meanings by asking purposeful questions and providing commentaries, and providing explanations concerning content and language quality (Varzari 2022).

Active communication during PhenoBL classes supports the communicative method of learning a foreign language. While analysing the phenomena, students not only focus on the grammatical and lexical quality of language usage but also on some relevant issues. In other words, learning activities are more focused on the message than on the form. Students and teachers collaborate in creating investigations, communicating, and creating new knowledge (Bobrowsky 2018; Fields 2018; Fields and Kennedy 2020; Johnson 2021; Bercasio and Adornado 2023).

While collaborating with their teachers, students overcome the spoken communication barriers, which are still determined by the deeply rooted, indoctrinated educational tradition, whereby a student is supposed to present the exact answer without making grammatical and stylistic language mistakes. Thus, the communicative activities of PhenoBL scaffold and further emancipate the development of spoken communicative skills in a foreign language (Ciuciulkiene and Stankeviciene 2014; Marsh et al. 2019).

These data complement the major experiential findings of teachers that highlight the increase in students' metacognitive awareness while applying the PhenoBL. Students with high metacognitive awareness know where and when to use knowledge (Wilson and Conyers 2016). Thus, it is only natural that PhenoBL is among the didactic priorities for those countries that are engaged in educational reform.

The topic of the implementation of phenomenon-based learning has hardly been studied in the context of educational practice in Lithuania. In individual studies, information has been presented on the few practices of several schools that have tried to integrate this method.

In 2021, a project took place in which the "School of Creativity", in cooperation with Tallinn University, who were invited to study 10 phenomena. However, this is more reminiscent of the first steps in trying to apply innovative methods rather than systematic practice. The same project identified 29 different disciplines integrating PhenoBL, but none of them was a foreign language. Presumably, the aim is not to teach a foreign language using PhenoBL. Although there is no unified and most effective way of teaching foreign languages, it is certainly known that, in order to learn a foreign language as best and as quickly as possible, it is important to be in an environment that speaks that language. It is

important, too, to not only listen but also speak that language and use it actively. Such a real-life, efficient simulation can be built using the PhenoBL paradigm.

The PhenoBL experience of Lithuanian foreign language teachers is analysed here in the context of the theoretical ideas and practical insights of PhenoBL teachers in foreign countries. PhenoBL has been applied as a new approach to learning (Tagunova et al. 2019), especially in teaching Russian as a foreign language (Makarova et al. 2020) and teaching English as a foreign language in Finland and Vietnam (Nguyen 2018), with PhenoBL also being implemented in Abu Dhabi (Valanne et al. 2017). Students learn together while forming a vocabulary relevant to their topic, develop it relying on each other's experiences. In this way, they actively teach each other and at the same time learn from each other, expanding their relevant vocabulary more intensively and polishing grammatical models.

The lack of research and experiential data on integrating PhenoBL in teaching foreign languages in Lithuanian schools makes this research relevant and innovative. The shortage of objective data presupposes the following series of research questions: Do teachers in Lithuania know what phenomenon-based education is? What is foreign language teachers' experience ofPhenoBL? What specific preparation do foreign language teachers need? These research questions model the further content of the research parameters.

The research itself is centred around the role of PhenoBL in teaching a foreign language. The main goal of this study is to analyse the experience of Lithuanian foreign language teachers in incorporating PhenoBL. Research data were collected while conducting semi-structured interviews. Semi-structured interviews allowed us to collect more detailed information about the investigation of PhenoBL in foreign language teaching. The data analysis method employed was qualitative inductive content analysis.

## 2. Materials and Methods

### 2.1. Research Design

The main aim of this research was to analyse the experience of Lithuanian foreign language teachers in incorporating PhenoBL. The concept of "experience" suggests the idea of a qualitative research paradigm (Mulisa 2022), allowing the researchers more space for data interpretation. The research data were collected while conducting semi-structured interviews. The data were analysed according to the model of inductive content analysis. The research was performed while following the ethical principles of qualitative research. The participants of the research were invited to share their experiences on a voluntary basis.

### 2.2. Materials and Procedures

To collect the main data, semi-structured interviews were organised. Semi-structured interviews were chosen due to the flexibility of this method and the possibility of collecting data from a group of 15 informants, inviting volunteer teachers from all Lithuanian schools that use PhenoBL. Semi-structured interviews also allowed the researchers to use more informative research questions that provided a clearer vision and understanding of the informants' experiences and individual approaches to the research situation. In this particular case, this situation constitutes the experience of foreign language teachers while implementing PhenoBL in their classes.

For the implementation of the semi-structured interviews, 23 main questions were prepared relying on an inductive approach (the operationalisation of the main theoretical concepts, relying on scientific analysis and lesson observation). This was based on the relevant theoretical and practical insights, which served as a premise for formulating the interview questions. The list of interview questions is presented in Table 1 below.

The theoretical analysis suggested three leading conceptual ideas concentrated around the experience of Lithuanian foreign language teachers who implemented PhenoBL in their curriculum. These ideas revealed the major meaning of PhenoBL, defining PhenoBL not only as a relevant, innovative, and timely learning strategy but also as a new way of thinking; its specification in teaching a foreign language frees up language usage and turns the foreign language into a successful tool for global socialisation, aligning with the special

didactic requirements for a foreign language teaching and combining language acquisition process domains (see Table 1). The general conceptual ideas served as a pretext for the research questions, which, in turn, were tied to the interview questionnaire.

**Table 1.** Operationalizsation of research and interview questions.

| Concept Content | Research Question | Interview Questions |
|---|---|---|
| The theoretically based PhenoBL concept states that PhenoBL stands for relevant, innovative, substantive, and timely learning. This method is not only a method but also a new way of thinking (Binkley et al. 2011; Symeonidis and Schwarz 2016; Lonka et al. 2018; Randolph et al. 2009; Akkas and Eker 2021; Roiha and Polso 2021; Bărbuleţ 2022). | Do foreign language teachers in Lithuania know what phenomenon-based learning is? | Did you have enough information about PhenoBL before implementing it into your language classes? |
| The relevant task of the foreign language teacher: to find the most effective strategies to ensure that a foreign language might become a tool, helping students to be successful in the global socialisation process (Bobrowsky 2018; Fields 2018; Protassova 2018; Fields and Kennedy 2020; Johnson 2021; Varzari 2022; Bercasio and Adornado 2023). | What is foreign language teachers' experience while facing PhenoBL? | What challenges do you face while using the PhenoBL method? What educational subjects are the easiest to integrate into the PhenoBL method? What academic subjects are the most challenging for the implementation of PhenoBL? What do you, as a teacher, lack in order to more smoothly incorporate PhenoBL into the curriculum for TFL? Do you notice a positive/negative impact on students' learning outcomes after starting to include this method? Comment your answer. What do you expect students to do before starting to explore the phenomenon? What is the attitude/mood of the students before/during/after investigating the phenomenon? How was the preparation process? How much time did you spend planning and how long did it take to successfully plan your investigation of the phenomenon? Is a 45 min lesson enough to apply PhenoBL? If not, how do you organise your class time? Do the students willingly engage in the study of the phenomenon? How did the learning process differ between less motivated and more motivated students? Did you involve parents in the process of researching the phenomenon? What basic knowledge did the students need to acquire before starting to study the phenomenon? Were students able to successfully integrate the necessary knowledge into the inquiry process? What are your recommendations for teachers who would like to try incorporating PhenoBL into their curriculum? Will you continue using this method? |
| Specific foreign language teachers' didacticskills, CLIL competencies (Ciuciulkiene and Stankeviciene 2014; Valanne et al. 2017; Nguyen 2018; Marsh et al. 2019; Tagunova et al. 2019; Makarova et al. 2020; Johnson 2021; Bercasio and Adornado 2023). | What specific preparation do foreign language teachers need? | Is collaboration between teachers an important part of the PhenoBL? Why? Do you think the inclusion of foreign languages in PhenoBL is important? Why? Do you include only the first foreign language or others as well? What methods and strategies does the language teacher apply while conducting PhenoBL? |

During the interview, the main interview questions were supported with clarifying questions. The research participants were allowed to develop their answers freely, i.e., if they wanted to speak further about certain issues. The questions were almost always asked in the pre-prepared order but adapted to the flow of the interview; if the respondent had already answered the question he wanted to ask, the question was not repeated. Sometimes, the researcher had to slightly clarify or rephrase the prepared questions. The duration of respondents' answers to the questions asked varied. The duration of a single semi-structured interview lasted about 1.5 h.

The interview was conducted by one researcher. The responses were recorded and later transcribed. The transcriptions were analysed by two independent researchers with the purpose of generating the main ideas and regularities. The main categories and subcategories were revealed. A validating consensus on the leading categories and subcategories was reached.

The semi-structured interview sample is based on purposive sampling: those teachers who are best able to answer the research questions based on their personal experience participate in the study. The chosen target group is teachers who are experts in education.

The research participants were selected according to two criteria: teachers who have already applied the phenomenon-based education method in their work; and those who teach not only in their native language but also include in the process a foreign language or languages. A total of 15 education experts were interviewed (n-15). Interviews were coded with a letter (M) and a number from M1 to M15.

The interviews were conducted until the researchers obtained "rich and thick" (Dibley 2011) data. Thick data means a lot of data; rich data are many-layered, intricate, detailed, nuanced, and more. One can have a lot of thick data that are not rich; conversely, one can have rich data that are sparse. For research validity, it is important to have both richness and thickness (Fusch and Ness 2015). This means that the interview texts (from 6 to 8 pages) were extensive and allowed for inductive content analysis.

The respondents participated in the survey while following the major principles of research ethics. The teaching experts participated in the survey voluntarily and their anonymity was ensured. During the interview, the teachers did not have to indicate their name, surname, or the name of their school. If the teachers gave the name of the school at which they worked, this was coded in the study. It is thus not possible to identify the respondents.

The received data were analysed using qualitative inductive content analysis according to Elo and Kyngäs (2008), while also relying on the insights of Lochmiller (2021). Qualitative inductive content analysis was performed, which is an inductive process involving iterative coding. By inductive process, we mean that the codes used to label the data are developed during the process of coding, based on the actual content of the data set. The codes were identified by the researcher within the data or, as is often said, as they "arise" within the data.

The received answers were analysed by three researchers using the MaxQda software. Finally, the received categories were validated by comparing the received results of the three researchers.

The major limitation of this study was the existing limited experiences of PhenoBl implementation in TFL. As it is still an innovation, the number of respondents who met the research requirements for active PhenoBL usage was quite low.

## 3. Results

After the analysis was performed, five leading major themes emerged: the competencies of student-centred teaching, the development of subject integration competencies, teamwork development competencies, major teacher achievements and challenges, and foreign language usage emancipation. It is worth noting that the teachers that participated in the research expressed more positive attitudes towards their experience of PhenoBL than negative. The revealed themes demonstrate that teachers paid major attention to the didactic possibilities of PhenoBL. Less attention was paid to the separate issues of evaluation systems.

The first theme was clarified from the answers to one of the semi-structured interview questions dealing with the foreign language teachers' primary knowledge of PhenoBL and their experience while applying it (see Table 1). The teachers highlighted the importance of student-centred teaching (see Table 2).

**Table 2.** Content of categories and subcategories of the first theme, "Competencies of student-centred teaching".

| Subcategory | Category |
| --- | --- |
| Good knowledge of students' interests, abilities, skills<br>Planning of students' experiences | Student-centred activity management |
| Independent choice of the phenomenon<br>Teacher as advisor and consultant | Student is the main path finder |

The evidence for the coding of this category coding can be illustrated by one of the responses: "<. . .> *Those who are more involved in this innovation often encourage other friends, set an example, etc. . .Teacher is not such a leader any more. There are students in the classroom who are different leaders than teachers. They know what they want. They have a different effect on those who are, say, quieter, than the teacher. The teacher has his own methods of encouragement, management and they have different methods of approach and encouragement among themselves*" *(M3).* Key ideas such as "*students are different leaders"* and "*they have different methods of approach and encouragement among themselves*" help to highlight teachers' understanding of the group, their study needs, and their ability to be their own learning leaders. This is why it is possible to speak about student-centredness in PhenoBL.

The second theme revealed the importance of subject integration competencies (see Table 3).

**Table 3.** Content of categories and subcategories of the first theme, "Development of subject integration competencies".

| Subcategory | Category |
| --- | --- |
| Holistic, integrative approach towards teaching<br>Knowledge about colleagues' interests | Collaboration among teachers |
| Good knowledge of curriculum<br>Creativity development<br>Time management | Phenomenon-compiling skills |

The evidence for category coding emerged from the following typical answers: "<. . .> *It is difficult to understand that my subject is not the most important. There is also no teaching of individual subjects, and the chosen phenomenon is studied, analysed, and produced through the prisms of many disciplines. Many educational subjects can be combined into one phenomenon, for example, when deciding to restore an old painting, mathematics, chemistry, art, a foreign language are included*" *(M1);* "*It is necessary to have good planning skills, to be willing to cooperate with colleagues—it is necessary to coordinate activities with other teachers, to search for additional information if the phenomenon involves disciplines that the teacher does not teach, to be creative*" *(M4; M7;* and *M14).* The categories were clarified while relying on the expressed didactic, collegial ideas of support, which allowed us to crystallise the major aspects of collaboration and a good knowledge of the phenomena curriculum.

The third theme stressed the relevance of the teamwork development competencies of teachers in enhancing the development of students' group-work competencies (see Table 4).

**Table 4.** Content of categories and subcategories of the third theme, "Teamwork development competencies".

| Subcategory | Category |
| --- | --- |
| Preparation of the phenomenon design in teaching groups<br>Flexibility of PhenoBL planning and organising | Team teaching |
| Dynamics of sharing responsibilities in the group<br>Collaboration of teachers and students | Development of students' group-work skills |

The presented answers "*Just as students preparing for research will work in a group, teachers could prepare and research in teams—this way they will have an even better understanding of the challenges that the students will have to face*" *(M7)*. "*There can be a study of 2–3 lessons, when the lessons of several teachers are combined, which are placed one after the other*" *(M4)* And

"*In the learning process, teachers are facilitators of learning, using their knowledge not necessarily to convey facts, but more importantly to encourage and guide students to solve problems that the students themselves have identified*" *(M13)*, allowed us to highlight one of the specific dimensions of PhenoBL—learning processuality—which is fulfilled with the help of team teaching.

Team teaching allows educators to turn ordinary class teaching into an authentic, problem-solving-based process, which is realised by turning groups of students into learning teams as well (Meriläinen and Piispanen 2012; Symeonidis and Schwarz 2016; Makarova et al. 2020). Becoming a learning team encourages inclusion and the development of communication and social skills.

The fourth theme was derived from answers to questions related to teachers' achievements and the challenges they faced while implementing PhenoBL (see Table 5).

**Table 5.** Content of categories and subcategories of the fourth theme, "Teachers' major achievements and challenges".

| Subcategory | Category |
|---|---|
| Children's enthusiasm and engagement in investigation<br>Feeling like a team member<br>Engagement in discussion in a foreign language<br>Range of didactic activities | Teaching achievements |
| Teachers' fears about making mistakes and misleading students<br>Discomfort of changing traditional teaching roles<br>Lack of administrative flexibility | Teaching challenges |

While speaking about their achievements, teachers once more highlighted a student-centred approach. An additional opinion was devoted to parents. The latter attitude supported innovative teaching and parental positiveness towards school. "*It's great fun when parents get involved. They are happy to see children's independent work. This is probably the greatest success, when the children's motivation grows, competencies improve. . . to see that even after the bell for a break, they are still discussing*" *(M12)*. "*I was happy to use my CLIL*" *(M9)*. "*That challenge is the same fear of saying something wrong. I myself feel like a student, because I learn together with the students, <. . .> This phenomenon is also a challenge, because the teachers do not know everything, and they are also researching something*" *(M10)*. As can be seen from the first citation, teachers are also mindful of their professional growth, reflect on emerging challenges, and evaluate the development of their competencies (Akkas and Eker 2021).

The last theme was derived from answers to questions dealing with the foreign language teaching methodical specification while using PhenoBL (see Table 6).

**Table 6.** Content of categories and subcategories of the fifth theme, "Foreign language usage emancipation".

| Subcategory | Category |
|---|---|
| Expansion of vocabulary<br>Usage of terms | Development of lexical competencies |
| Better usage of interrogative sentences<br>Better usage of negative sentences | Polishing of grammar models |
| Value-based speaking (speaking the truth)<br>Presenting arguments<br>Courage for public speaking | Development of oral communication skills |

While talking about foreign language learning from the perspective of PhenoBL research, participants stressed the importance of the development of creativity, collaboration, and communication: *"Public speaking skills are very important. After completing the research, it needs to be presented, so students must not only be original and creative, but also accurate, good time managers. It means that they have to make a speaking plan, prepare a presentation text, also be able to follow that plan. Especially if the presentation is carried out in a foreign language, in front of an audience, children learn to overcome their fears and clarify areas in which they could do some extra work. Especially amazing to see them emotionally involved in their friends' speaking, becoming supportive, collaborative fans"* (M5). This attitude almost coincides with researchers' attitudes that learners are constantly encouraged to actively participate in researching and analysing specific academic information, in reflecting on emerging challenges with peers and groups, in debating and negotiating, in drawing conclusions, obtaining results, and reflecting on their experiences throughout the learning process (Roiha and Polso 2021).

After carrying out a qualitative study and interviewing teachers already using this method, it became clear that no special preparation is needed for this method to be applied; the most important thing is that teachers are theoretically familiar with the application of this method, are interested in examples and good practices, have the know-how, and are willing to cooperate with colleagues in order to share their acquired experiences. It is important that the school administration tend toward being more flexible regarding temporary changes in the educational process, but for the method to be successfully applied, this condition is not mandatory; teachers are able to apply the PhenoBL method while adapting to the existing norms in the school.

## 4. Discussion and Conclusions

The success of the PhenoBL method is greatly influenced by the preparation of teachers for the implementation of this innovation, by the possibility of involving experts, and by the school's flexibility in temporarily adapting the schedule, providing the opportunity to change the learning space. This can be supported by the scientific research data highlighting the importance of the pedagogical environment becoming accessible, concrete, and meaningful to learners (Lonka et al. 2018; Roiha and Sommier 2018; Akkas and Eker 2021; Silander et al. 2022). The research participants (Lithuanian teachers) also highlighted the importance of didactic contexts that develop students' metacognitive awareness and stimulate their curiosity and creativity.

As Lithuanian teachers state, the uniqueness of the phenomenon method is that it goes beyond the boundaries of one educational subject—the research aims to create the conditions of a real-life problem and teach students to solve real problems. The ability to integrate this with other acquired skills (public speaking, global perspectives, etc.) will benefit students after school, as they can (and probably must) be used outside of school. The latter approach supports the attitude about the holistic nature of PhenoBL, linking practical and realistic life topics into a unique whole, forming a new way of thinking about learning (Roiha and Sommier 2018; Akkas and Eker 2021; Silander et al. 2022).

The phenomenon-based approach brings students closer to the real world and enables them to solve real-world problems while communicating in a foreign language in a safe and error-promoting environment. Teachers who use PhenoBL in their classes are positive, highlighting the benefits it brings to both teachers and students.

All teachers tend to recommend this method to colleagues who have not yet tried it, encouraging them to not be afraid to make mistakes and learn from them. However, this method also has challenges that can scare educators; it is important to cooperate with each other regardless of teaching subject by dividing the workload and saving time, sharing knowledge, reading a lot, and being interested before applying PhenoBL. The most widely discussed topic is how teachers and students incorporate foreign languages into the study of the phenomenon or teaching a foreign language using PhenoBL.

The intensity of language activities improves the understanding of a foreign language and reduces the fear of public speaking. In addition, the teachers distinguished that the

way students use foreign languages is related to their age and experience—older students who know foreign languages better prefer to communicate in a foreign language.

**Author Contributions:** Conceptualisation, N.C.; methodology, I.T.-B.; software, M.C.; validation, I.T.-B.; formal analysis, N.C., I.T.-B. and M.C.; investigation, N.C.; resources, M.C.; data curation, N.C. and I.T.-B.; writing—original draft preparation, N.C.; writing—review and editing, I.T.-B.; visualisation, I.T.-B.; supervision, I.T.-B.; project administration, N.C.; funding acquisition, M.C. All authors have read and agreed to the published version of the manuscript.

**Funding:** The research was financed by the Council of Lithuanian Academy of Sciences (project registration number: P-SV-22-43). Project title: Analysis of the experience of implementing the phenomenon-based method (PhenoBL) of Lithuanian pedagogues in teaching foreign languages July 2022–October 2022.

**Institutional Review Board Statement:** This research was carried out following the provisions that underline the basic principles of professionalism and the ethics of research and approved by Resolution No. SEN-N-17 of the Senate of Vytautas Magnus University on 24 March 2021. A separate decision by the Ethical committee was not required as we used the open data of the research, which was financed by the Council of the Lithuanian Academy of Sciences (project registration number: P-SV-22-43). Project title: Analysis of the experience of implementing the phenomenon-based method (PhenoBL) of Lithuanian pedagogues in teaching foreign languages July 2022–October 2022.

**Informed Consent Statement:** Informed consent was obtained from all subjects involved in the study.

**Data Availability Statement:** No new data were created or analyzed in this study. Data sharing is not applicable to this article.

**Conflicts of Interest:** The authors declare no conflict of interest.

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
