# Peer review of "Phenomenon-Based Learning in Teaching a Foreign Language: Experiences of Lithuanian Teachers"

_socsci, doi:10.3390/socsci12120670_

Round 1

Reviewer 1 Report (Previous Reviewer 1)

Comments and Suggestions for Authors

Comments on revised version:

• Framework (literature study) in the field of language teaching/learning could be expanded
• Hypotheses that compare the chosen method to other methods are rather implicit
• Motivation of the questionnaire should be explained in a more detailed way.  
• The “foreign language” component in the study could be more underlined and lead to interesting recommendations for teacher training.

Author Response

  1. Framework (literature study) in the field of language teaching/learning could be expanded - Expanded with 12 references;
  2. Hypotheses that compare the chosen method to other methods are rather implicit - Hypothesis are not obligatory in a qualitative research. We present research questions;
  3. Motivation of the questionnaire should be explained in a more detailed way - motivation is presented in expended Table 1.
  4. The “foreign language” component in the study could be more underlined and lead to interesting recommendations for teacher training - The component of the foreign language teaching is expanded with the theoretical insights of language teaching dimensions, structural aspects of PhenoBL that enable different communicative activities.

Reviewer 2 Report (Previous Reviewer 2)

Comments and Suggestions for Authors

In the abstract it is recommended to eliminate the words that give name to different sections of the research: Research objective, Search methods, Research results, etc.

The Introduction is very short. It must be expanded. There is much more bibliography that is not cited and that should appear on the topic discussed. Furthermore, it is current and pertinent to mention it. The Introduction section should serve so that the reader can get an idea of the current reality on the topic discussed and as it stands, this idea cannot be created.

Research funding should not appear where it does in this article: at the end of the Introduction section, but rather it should appear at the end of the article, in an Acknowledgments section.

Section 2: Materials and methods, contains about four initial paragraphs that should be within a subsection indicating that it corresponds to the part of the research in which the scientific literature is analyzed. And the rest of the paragraphs of the section should also go in another subsection where qualitative research is discussed. Both subsections should be preceded by a short introductory text, warning the reader of the two parts of the research.

At the end of section 2, some limitations of the study are mentioned. It is not the place where they should be, but rather they should go in the Conclusions section, along with other limitations that may exist both in the bibliographic search and in the results. One of these limitations to add is the limited sample (15 teachers) that the research has.

The conclusions seem to me to be excessively descriptive and not very persuasive regarding the achievement of the stated objectives. The wording of this section could be polished and the central ideas given as conclusions could be improved.

Likewise, as already indicated, the limitations of the study, which are several, and the future lines of research that they intend to carry out must also be added.

References are current and relevant. But as has already been mentioned, the Introduction section needs to be expanded, and more references will surely come from there.

Those that exist, although as has been said, are current and relevant, are very few. It should be expanded with more references.

Author Response

  1. In the abstract it is recommended to eliminate the words that give name to different sections of the research: Research objective, Search methods, Research results, etc. - The abstract content is changed according the recommendations.
  2. The Introduction is very short. It must be expanded. There is much more bibliography that is not cited and that should appear on the topic discussed. Furthermore, it is current and pertinent to mention it. The Introduction section should serve so that the reader can get an idea of the current reality on the topic discussed and as it stands, this idea cannot be created. - The introduction is expanded. The communicative language while using the PhenoBL is highlighted. The list of references is expanded with 12 positions. The authors concentrate on the material focusing on communicative approach while teaching a foreign language.
  3. Research funding should not appear where it does in this article: at the end of the Introduction section, but rather it should appear at the end of the article, in an Acknowledgments section. - Placed at the recommended position.
  4. Section 2: Materials and methods, contains about four initial paragraphs that should be within a subsection indicating that it corresponds to the part of the research in which the scientific literature is analyzed. And the rest of the paragraphs of the section should also go in another subsection where qualitative research is discussed. Both subsections should be preceded by a short introductory text, warning the reader of the two parts of the research. - It is corrected according the recommendations.
  5. At the end of section 2, some limitations of the study are mentioned. It is not the place where they should be, but rather they should go in the Conclusions section, along with other limitations that may exist both in the bibliographic search and in the results. One of these limitations to add is the limited sample (15 teachers) that the research has. - The Limitations place is not fixed. The researchers decided to leave it in a methodological part as it belongs to the research sample and procedures evaluation.
  6. The conclusions seem to me to be excessively descriptive and not very persuasive regarding the achievement of the stated objectives. The wording of this section could be polished and the central ideas given as conclusions could be improved. - Corrected.
  7. Likewise, as already indicated, the limitations of the study, which are several, and the future lines of research that they intend to carry out must also be added. - Corrected.
  8. References are current and relevant. But as has already been mentioned, the Introduction section needs to be expanded, and more references will surely come from there.  - Expanded with 12 references. The introductory part is expanded. 
  9. Those that exist, although as has been said, are current and relevant, are very few. It should be expanded with more references. - Expanded with 12 references.
  10.  

This manuscript is a resubmission of an earlier submission. The following is a list of the peer review reports and author responses from that submission.

Round 1

Reviewer 1 Report

Comments and Suggestions for Authors

Is the content succinctly described and contextualized with respect to previous and present theoretical background and empirical research (if applicable) on the topic?

The author should integrate a state of the art on pedagogical approaches other than

 Phenomenon-based learning (PhenoBL). As it stands, the article gives the impression that there is no activating approach beyond that described in the research.

The author should also broaden his framework to the teaching and learning of foreign languages and much more present the parameters/variables that influence the success of this teaching/learning.

Are all the cited references relevant to the research?

The references are relevant but incomplete. This follows from the limitations I presented above. No allusion is made to the theories concerning casus based learning, task based learning, problem based learning.

In addition, it would have been interesting to integrate more research that analyzes the effect of simulation-based instruction, simulation which is becoming more and more important nowadays with the possibilities of the web.

Are the research design, questions, hypotheses and methods clearly stated?

The author lists the research questions, without accompanying them with hypotheses that could have arisen from a study of the scientific literature.

In order to collect data, the author used a questionnaire during interviews. We would have liked to have this questionnaire. The questions now appear in a fragmented way in the text.

Are the arguments and discussion of findings coherent, balanced and compelling?

The author cites a number of responses collected and summarizes them in clear tables. These answers are then briefly commented. However, we would have liked to have had a more detailed analysis of the way in which teachers integrate the approach, of the materials (authentic documents of what nature?), of the work forms and of the evaluation methods.

In addition, the reader wonders what are the objectives that teachers want/must achieve with their students and how they try to achieve this with the method studied (e.g. flipped classroom?).

The style of the abstract and the end of the introductory part is too fragmented and the author should also adopt a more neutral tone. The reader has the impression of attending from the beginning a plea in favor of PhenoBL.

For empirical research, are the results clearly presented?

The results are clearly presented but should be more consistent. The method studied undoubtedly implies an approach based on interdisciplinarity, which could have given rise to some critical reflections.

Is the article adequately referenced?

No remarks

Are the conclusions thoroughly supported by the results presented in the article or referenced in secondary literature?

Since the article is incomplete, so are the conclusions.

Tips for improving findings:

• Expand the framework (literature study), both in terms of pedagogical approaches (the one chosen and other well-known approaches) and in the field of language teaching/learning

• Formulate research questions, accompanied by hypotheses that compare the chosen method to other methods

• Show how the questions posed to participants are based on previous studies and to what extent certain questions are original, or adapted to the specific context of the study; take into account all the aspects necessary for the development of a language course

• Develop much more the “foreign language” component in the study

These few tips should make it possible to reach more interesting conclusions and to formulate recommendations for teacher training.

Author Response

The main concept of PhenoBL was expanded with the notes on Content based learning, task based learning while comparing them to problem based learning, inquiry based learning, PhenoBL. It is worth to mention that in our research teachers were talking about real life situations the analysis of which is important for PhenoBL. The list of references was expanded by new author who we considered to be suiteble for the answering the reviewer's notes. See reference: Gabriel, Bărbuleţ. 2022. Content based learning-Task based learning-Problem based learning in Teaching Romanian Language to Foreign Students. For the qualitative research the hypothesis is not compulsory, so we pose the problem questions: do teachers in Lithuania know what phenomenon-based education is? What is their experience while facing the Pheno BL? How can we help schools prepare to integrate PhenoBL into their curriculum? What specific preparation do foreign language teachers need? .The interview questions are included in the text. It is worth to mention that the revealed themes demonstrate that teachers paid major attention to didactic possibilities of PhenoBL. Less attention was paid to the separate issues of evaluation systems. 

Reviewer 2 Report

Comments and Suggestions for Authors

The introduction section is too short. It should be expanded and more quotes should be incorporated to support the arguments that are given regarding the subject matter.

Put the thanks to the financing to do this research at the end of the article.

It should be better explained how the interviews and the analysis of the responses to the questions were conducted. One or more researchers, audio recorded and later transcribed, etc.

It is not indicated which software was used to do the qualitative analysis of the answers given in the questions (Atlas.ti, MaxQda, Excel, etc.)

The number of citations and references is quite low. The number of citations should be increased, incorporating impact and current citations. Although they are studies focused on European countries, they are of interest and should appear.

In the conclusions, the limitations of this case study should be mentioned: an intentional and very small, non-representative sample.

Author Response

The introduction is expanded by comparing the PhenoBL with other didactic strategies such as content based learning, task based learning, PBL, inquiry based learning. The acknowledgements are presented at the end of the research paper. The methodological part is expanded with the procedures explaining the conducting of the interview and received data analysis: lines 184-186 and 203-206.  As the amount of the text is limited we expanded the introduction and theoretical part including one more source of references (Gabriel, Bărbuleţ. 2022). Limitations are included as well see Line 208-211. 

Round 2

Reviewer 1 Report

Comments and Suggestions for Authors

This revised version gives the reader a more fine grained analysis and could be of interest for other scholars.